# Cancer gene mutation frequencies for the U.S. population

Gaurav Mendiratta [1,3✉], Eugene Ke [2,3✉], Meraj Aziz [1,3], David Liarakos[1], Melinda Tong[1] & Edward C. Stites [1✉]

Mutations play a fundamental role in the development of cancer, and many create targetable vulnerabilities. There are both public health and basic science benefits from the determination of the proportion of all cancer cases within a population that include a mutant form of a gene. Here, we provide the first such estimates by combining genomic and epidemiological data. We estimate *KRAS* is mutated in only 11% of all cancers, which is less than *PIK3CA* (13%) and marginally higher than *BRAF* (8%). *TP53* is the most commonly mutated gene (35%), and *KMT2C*, *KMT2D*, and *ARID1A* are among the ten most commonly mutated driver genes, highlighting the role of epigenetic dysregulation in cancer. Analysis of major cancer subclassifications highlighted varying dependencies upon individual cancer drivers. Overall, we find that cancer genetics is less dominated by high-frequency, high-profile cancer driver genes than studies limited to a subset of cancer types have suggested.

[1] Integrative Biology Laboratory, Salk Institute for Biological Studies, La Jolla, CA 92037, USA. [2] Department of Microbiology, Immunology, and Cancer, School of Medicine, University of Virginia, Charlottesville, VA 22903, USA. [3] These authors contributed equally: Gaurav Mendiratta, Eugene Ke, Meraj Aziz. ✉email: gmendiratta@salk.edu; ek7kn@virginia.edu; estites@salk.edu

It has long been recognized that sequencing the genomes from cancer patients would identify those genes relevant to human health and disease[1]. Genomic sequencing of DNA from cancer patient tumor samples has now characterized the relative abundances of mutations for all genes and for many different forms of cancer[2–7]. The identification of commonly mutated genes has been important for drug development because several oncoproteins have proven to be valuable drug targets[8–14].

A better understanding of which genes are most commonly mutated across all cancers, and at what frequency, could help prioritize genes and pathways in a manner that increases public health benefits. For example, the RAS oncogenes are widely believed to be mutated in ~30% of all human cancers. The National Cancer Institute created its Ras Initiative to focus upon RAS because of the perceived, high, unmet burden of RAS mutant cancers[15]. Along these lines, a better understanding of the burden from known cancer drivers may help better prioritize research funding.

An improved understanding of how common different mutations are in cancer patients could also benefit drug development and personalized medicine. Clinical trials are increasingly employing a basket-trial format[16,17] in which mutated forms of a protein are treated equivalently across different types of cancer. For example, larotrectinib and entrectinib both received FDA approval, in part, on the basis of basket-trial data that demonstrated efficacy across a variety of tumor types[17–19]. Additionally, pembrolizumab received FDA approval based on basket-trial data[17,20,21]. As FDA approvals may be increasingly based upon mutation in a cancer-type agnostic manner[18,20], the benefit of finding mutations that are collectively common across different tissues increases. Progress in personalized medicine is also being made through the off-label utilization of FDA-approved agents in attempts to inhibit targetable variants in forms of cancer other than those specified in the current approval(s)[22–25]. An improved characterization of the overall prevalence of mutations in targetable genes may help with the design of such personalized medicine clinical trials.

Although the proportion of patient tumor samples with a gene mutation is readily available for many specific types of cancer and also from multiple pan-cancer studies that combine samples from a variety of cancers[26–28], the field does not yet have estimates for the overall percentage of cancer patients in a population that harbor a mutated form of a specific gene. That accurate mutation prevalence values across all cancers are not available may be surprising at first glance because it may seem that this information is readily available in resources like cBioPortal[29,30], COSMIC[31], TCGA[26,27], AACR GENIE[32], and MSKCC impact[33]. Although several of these resources can provide a value for the percentage of samples with a mutation in a given gene, the representation of different cancers in these resources is not designed to be proportional to the relative burden of those cancers within the population.

If the frequencies of mutations for each gene within individual types of cancer were weighted by the relative abundances of each cancer it would be possible to calculate the overall proportion of cancers within a population that have a gene mutated. (Note: we will use the term mutation proportion to refer to this population-level probability of observing a mutation within a cancer patient's tumor.) The National Cancer Institute (NCI) Surveillance, Epidemiology, and End Results (SEER) Program has tracked cancer incidence within the USA since the mid 1970's[34]. These data are utilized in the annual cancer epidemiology survey reported by the American Cancer Society[35]. However, it is not possible to simply weight mutation frequencies from cancer genomics studies by the number of cases of that same cancer observed epidemiologically due to a lack of consistency in cancer classification system

utilization in genomics and epidemiology (Fig. 1). SEER characterizes each cancer by two International Classification of Diseases for Oncology, third revision (ICD-O-3) codes[36]. One specifies the anatomical location of the site of origin, and the other specifies the tumor histology, and many cancers require both to be uniquely classified. In contrast, many cancer genomics studies do not provide ICD-O-3 codes for their samples but rather describe their pathological diagnosis with broad general terms, such as breast cancer or pancreatic adenocarcinoma.

Here, we calculate and present the estimated proportion of all cancers that harbor one or more mutations within each gene of the genome. We do this by integrating epidemiological and genomic data. We overcome the obstacle created by a lack of consistency in cancer classification systems by developing and then implementing a process for mapping between these two data types. Our results reveal that specific cancer driver mutations are much less common than believed, with several high-profile oncogenes being much less common than routinely stated.

## Results

**Development of ROSETTA.** In order to integrate cancer genomic data with SEER epidemiological data, we developed an approach to interconvert between the different nomenclatures used to label cancer diagnoses. We did this by creating ROSETTA (Reclassification Of Sequencing and Epidemiological Tumor Type Annotations). ROSETTA is effectively an alternative classification system that maps to each of the other two classification systems, and it thereby enables the integration of epidemiological and genomic data (Supplementary Document 1; Supplementary Software 1). ROSETTA categories were created by grouping similar ICD-O-3 classifications within a single category, just as a cancer genomics study may focus on one general form of cancer (i.e., lung adenocarcinoma) where the collection of samples pools from 25 or more of the detailed ICD-O-3 adenocarcinoma subtypes that are utilized in epidemiological records.

Our ROSETTA categories have both biological and pragmatic influences. Groupings are based on biological factors that are already incorporated into the overall organization of ICD-O-3 codes. The granularity of a grouping is also influenced by choices that were made in previous cancer sequencing studies. For example, ROSETTA categories map broadly onto SEER categories when the corresponding sequencing studies are very broad and cannot be subdivided on the basis of metadata. When genomic studies can be further subdivided on the basis of included metadata, multiple ROSETTA classification terms may be utilized for a single genomics study to increase the resolution of the mapping between data sets. Our approach results in 370 different cancer classification categories of varying levels of granularity. The mixed granularity with how finely cancers are subdivided in ROSETTA suggests that the number of ROSETTA categories should not be used as a metric of the number of types of cancer at any one level of detail but should rather be considered a metric of the scale of this tool that enables genomic and epidemiological integration.

**Reclassification of epidemiological data by ROSETTA.** We utilized SEER epidemiological data that tracks malignant cancer diagnoses between the years 2000 and 2017, with a regional distribution that covers approximately 25% of the US population[34] (Fig. 2a). We considered the proportions of cancers diagnosed in this population to be a reasonable estimate of the proportions of cancers diagnosed across the entire US population; supporting our assertion is that the annual ACS epidemiological estimates for nationwide cancer incidence[35] correlate very well with these data, with Pearson's $r = 0.98$ and $p < 0.00001$ (Fig. S1). Overall, this SEER dataset

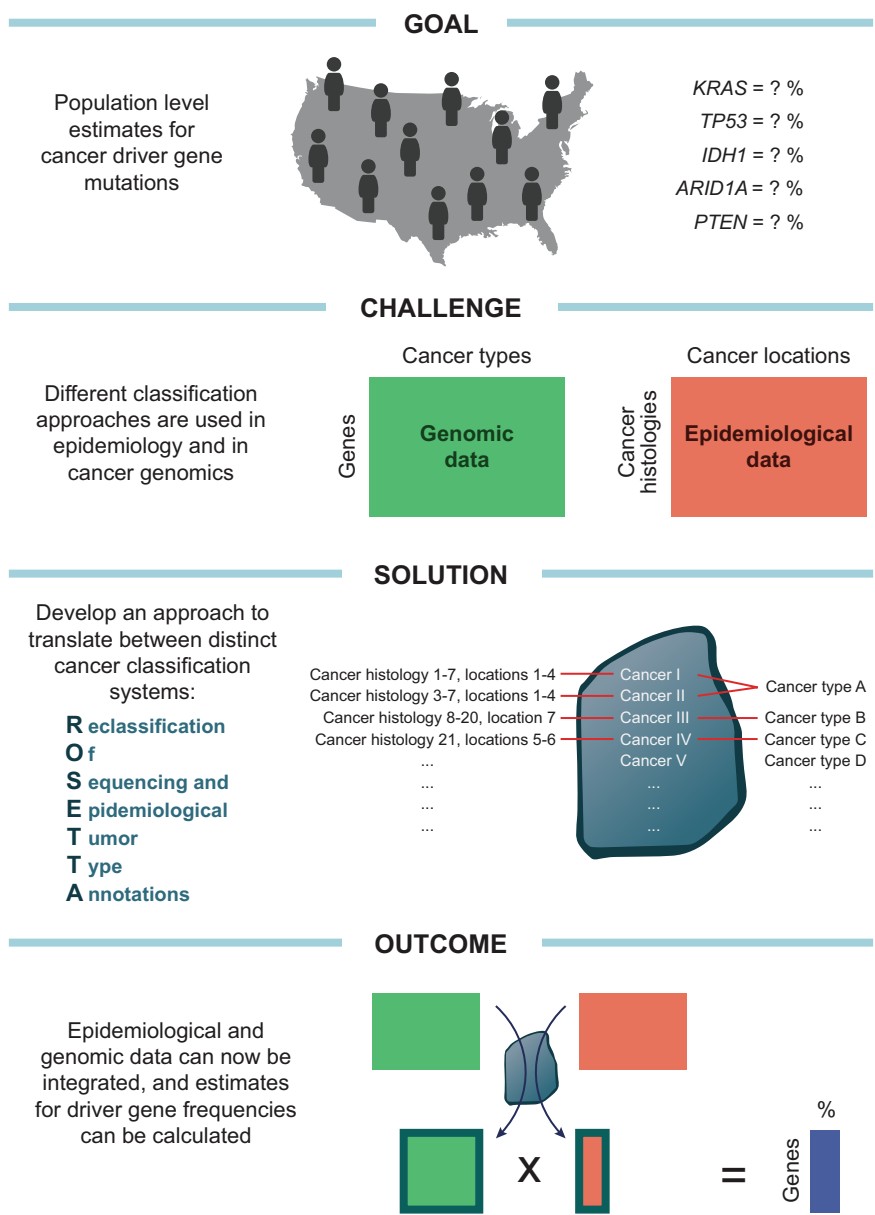

**Fig. 1 Approach for calculating epidemiologically weighted cancer gene mutation proportions across a population.** The major obstacle for combining genomic data with epidemiological data is that each utilizes a distinct cancer classification system. To overcome this obstacle, we developed a reclassification (ROSETTA) that could be a common format for both cancer genomic data and cancer epidemiological data and thereby enable mutation frequencies to be directly calculated.

comprises more than seven million different cancer diagnoses, with each diagnosis specified by one of 670 ICD-O-3 malignant cancer histology (morphology) codes and by one of more than 100 ICD-O-3 site (anatomical location or topography) codes (Fig. 2a). We reclassified the histological diagnoses from epidemiological studies in accordance with ROSETTA and performed quality control checks at different intervals of processing to ensure the validity of our data processing (Fig. S2).

After ROSETTA reclassification, we found the total cancer incidence for each cancer and formatted these values as a vector that provided the proportion of all cancers within each ROSETTA category (Supplementary Data 1). The five most common cancer types were breast carcinoma, prostate cancer, colorectal adenocarcinoma, lung adenocarcinoma, and malignant melanoma, which together comprised more than 50% of all

cancer diagnoses. The ten most abundant categories comprised 68.7% of all cancer diagnoses. Of note, some pan-cancer studies do not include samples from each of the ten most abundant types of cancer[26,37].

**Reclassification of genomic studies by ROSETTA and pooled analysis.** For cancer sequencing data, we included exomes from 19,181 cancer patients that were a part of 139 different cancer sequencing studies (Fig. 2a). We obtained mutation calls for exome-based genomic studies that were previously performed on samples from human cancer patients. A complete list of studies utilized is provided in Supplementary Data 2. We manually curated and assigned ROSETTA codes to sequenced samples on the basis of the attached metadata that revealed the finer histopathology of the sample (Fig. S2). We then automated the

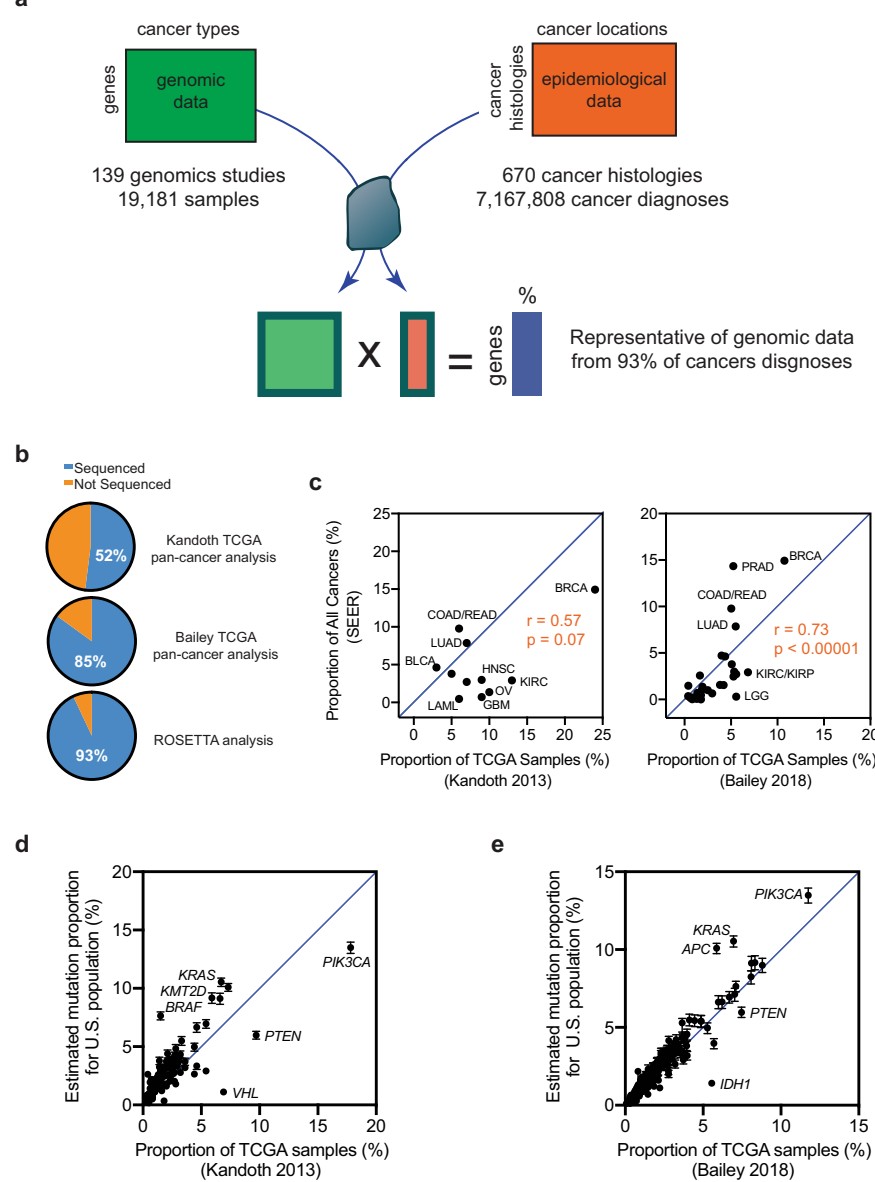

**Fig. 2 Properties of the data analyzed. a** We enable integration of cancer genomic and epidemiological data by implementing the ROSETTA cancer classifications. **b** Comparison of the proportion of all epidemiological cancer incidences for which representative genomic sequencing data is available in previous TCGA pan-cancer analyses and the present study. **c** Comparison of the proportion of TCGA pan-cancer datasets attributed to specific cancer types with the epidemiological proportion for the same cancer types, along with the Pearson correlation and *p*-value (2-tailed) (*p* = 0.07 for left panel, *p* < 0.00001 for right panel). **d** Comparison of the weighted gene mutation proportion from the present study with the mutation incidence from the Kandoth et al. TCGA pan-cancer analysis. **e** Comparison of the weighted gene mutation proportion from the present study with the mutation incidence from the Bailey et al. TCGA pan-cancer analysis. In panels **c–e**, the *x* = *y* diagonal is presented to highlight deviations between the two data sets. Error bars in panels **d** and **e** represent the 95% confidence intervals, *y*-coordinates represent calculated mutation proportions and *x*-coordinates are data points from literature with no error estimates available.

recording of which genes were mutated at least once with either a missense mutation, nonsense mutation, or short indel for each sample, maintaining these counts within each sample's assigned ROSETTA classification. As our goal was to tabulate the proportion of cancer patients who have a gene mutated (or not), we counted the patient sample as mutated whether it had a single mutation or multiple mutations within that same gene. When multiple samples from the same patient had been sequenced, we included only the original sample in our analysis. Further, some cancer genomics studies aggregate previously published samples along with their new samples. To avoid double counting of the

same sample, we identified overlapping samples between studies on the basis of patient and sample identification classifiers, and we only included mutation calls from the most recent study in our analysis.

There are a wide variety of ways that a gene can be mutated, including mutations both to intronic and exonic regions[38,39]. In our present analysis, we focus on mutations that alter gene coding and can be reliably detected at a high rate in exome data, including missense, nonsense, and indel mutations. We highlight that our use of the term mutation in this study is limited to these types of mutations, and that we are not including gene fusions,

intronic (e.g., splice site) mutations, or copy number mutations (i.e., amplifications or deletions). Now that we have developed and presented ROSETTA, our same approach for integrating epidemiological data with genomic data could be applied to other types of genomic data and to other forms of mutations in future studies.

**Evaluation of cancer-type coverage and comparison with previous pan-cancer studies.** With both SEER and cancer genomics re-classified according to ROSETTA, it was possible to integrate these data. We estimated the overall proportion of mutations to any gene within the cancer patient population as the sum of products of the gene mutation proportion in a cancer type with the percentage of all cancer incidences that is due to that cancer type (Fig. S2).

Even though there are many different types of cancer, the more common forms of cancer make up a disproportionately large fraction of all cancers. We found that we had sequencing data representative of 93% of cancer cases (Fig. 2B). In our estimates provided here, we consider the epidemiologically weighted mutation proportion from this 93% of diagnoses to approximate the mutation frequencies for the remaining 7% of cancer diagnoses for which genomic sequencing data is not readily available. We note that a similar but stronger assumption is implicit to other pan-cancer and pooled analyses that do not weigh for epidemiological rates of cancer and that include a smaller percentage of cancer types. We also note that further sequencing of the additional 7% of cancer diagnoses will eliminate the need for this assumption, and will reveal whether this assumption had a biologically meaningful impact on overall estimates.

We compared our weighted mutation proportions to two (unweighted) TCGA pan-cancer analyses of gene mutation incidences. The first pan-cancer TCGA analysis by Kandoth et al.[26] had representative sequencing data for 52% of cancer cases (Fig. 2b). The number of samples per cancer had poor proportionality to overall cancer frequencies (Fig. 2c). We compared mutation frequencies of the 125 protein-coding genes for which the Kandoth et al. analysis concluded were cancer driver genes and also provided pan-cancer mutation frequencies[26]. Although there is good general agreement between the proportions of samples with mutations in their dataset and our epidemiologically weighted estimates, a comparison of each study's results revealed that one would be making an error if one considered the unweighted pan-cancer analysis to be representative of the mutation proportions in the general cancer patient population. For example, the unweighted studies found *VHL* and *BRAF* mutations incident at 7 and 2%, respectively, while our weighted study estimates these genes to be mutated in 1 and 8% of the cancer population. These differences most likely follow from the fact that the TCGA pan-cancer dataset was not designed to be proportional to cancer incidence, and cancers with these mutations were over-represented and underrepresented, respectively, relative to other cancer types (Fig. 2d).

The more recent TCGA pan-cancer analysis by Bailey et al.[27] had representative sequencing data for 85% of cancer cases (Fig. 2b), and the number of sequenced samples of each cancer was a more proportionate representation of cancer diagnosis frequencies (Fig. 2c). We compared mutation frequencies of the 299 genes that the Bailey et al. analysis concluded were cancer driver genes and also presented pan-cancer mutation frequencies[27]. Although there is good general agreement between the unweighted pan-cancer mutation frequencies and our epidemiologically weighted mutation proportions, many individual genes had mutation rates that differed between the approaches. For example, this unweighted study found *KRAS*, *APC*, and *IDH1* mutations at frequencies of 7, 6, and 6%, respectively,

while our weighted study finds them to be mutated at rates of 11, 10, and 1%, respectively (Fig. 2e). Thus, although the Bailey et al. study had improved cancer-type coverage and improved relative proportional representation, considering these values as estimates for the actual mutation frequencies could still lead to relatively large errors for some commonly mutated genes.

Of note, there is no gold standard against which we can compare our ROSETTA-based estimates. However, pan-cancer gene mutation frequencies would be anticipated to be better estimates for true frequencies as the coverage of cancer types and the proportional representation of different cancers improve. Such a trend is seen between these pan-cancer studies and our ROSETTA-based estimates. This provides some additional validation for our methodology and its implementation.

**Cancer incidence mutation frequency estimates.** We focused on the calculated population-level mutation proportions for genes in the COSMIC Cancer Gene Census list (Tier 1) (Fig. 3a; Table 1; and Supplementary Data 3)[31]. *TP53* was the most commonly mutated cancer driver gene, which we estimate to be mutated in 35% of new cancer diagnoses. Even though *KRAS* is widely believed to be the most commonly mutated proto-oncogene[40–42], our epidemiologically weighted estimates found *KRAS* mutations (11%) to be less common than *PIK3CA* mutations (13%). The *BRAF* oncogene was found to be mutated in 8% of cancers, which is only marginally less common than *KRAS*. Three epigenetic modifiers (*KMT2C*, *KMT2D*, and *ARID1A*) were among the ten most frequently mutated cancer driver genes, highlighting the frequency of epigenetic dysregulation in cancer[43]. Overall, it was surprising that the most frequently mutated cancer-associated genes were mutated in only a small proportion of all cancers. As our analysis includes both driver and passenger mutations and is simply looking at the total number of mutations found within the genes in the Cancer Gene Census, this suggests that the overall occurrence of driver gene mutations in human cancer is even less than the values we present here.

Protein kinases with pathogenic mutations have proven to be valuable drug targets for which multiple FDA-approved small molecule inhibitors have been developed[8–14,44]. We therefore, considered the overall mutation frequency across all protein kinases (Fig. 3b, Supplementary Data 3). We found the two most commonly mutated kinases to be *TTN* (30%) and *OBSCN* (9%). *TTN* and *OBSCN* encode two of the largest proteins to have a kinase domain (approximately 34,000 and 8,000 amino acids, respectively), and the large size of these genes likely contributes to their high frequency of mutation[45]. Although there may be some theoretical arguments that a fraction of the mutations in very large genes may not be passenger mutations[46], mutations in these kinases are commonly assumed to be passenger mutations, and these two genes are not included on the COSMIC Cancer Gene Census list[31]. In contrast, *BRAF* and *ATM* are both well-established cancer driver genes, and these two kinase-containing genes were found mutated in 8 and 5% of cancers, respectively. All other protein kinases were estimated to be mutated in < 5% of all cancers. That kinase genes like *BRAF* and *ATM* with strong selective pressure for a cancer-promoting mutation would be so much less commonly mutated than large genes with no (or low) selective pressure may suggest that each driver gene's selective pressures is not present in every tissue, and/or may suggest that the selective pressure is influenced by co-occurring mutations. Again, we note that the mutation proportions we calculated are for all mutations and are not limited to those mutations most likely to be pathogenic. Additionally, it is important to note that not all mutant forms to a driver gene kinase are equally targetable[47].

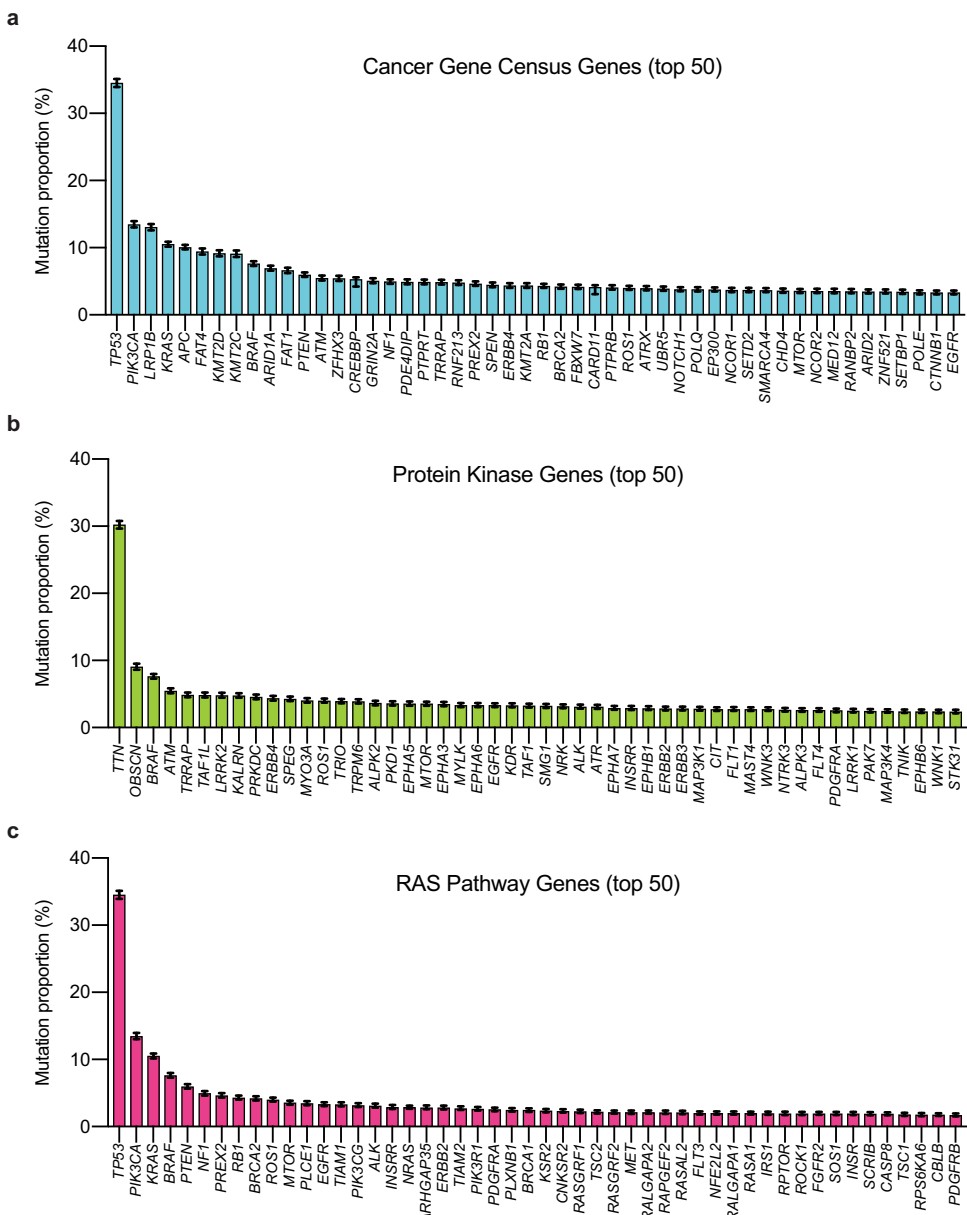

**Fig. 3 Epidemiologically weighted gene mutation proportions for sets of genes important to cancer. a** Estimated cancer patient population mutation proportions for the fifty most mutated genes from the tier one genes of the Cancer Gene Census. **b** Mutation proportions for the fifty most mutated protein kinase genes. **c** Mutation proportions for the fifty most mutated genes from the NCI Ras Initiative Pathway 2.0 gene set. Mutation proportions for all genes in each class are provided in Supplementary Data 3. Error bars in all panels represent the 95% confidence intervals determined through simulated samples ($n = 2,000$ independent Poisson distributed computational samples for each gene and histology with the calculated mutation proportion as the central value), and bar levels represent calculated mutation proportions.

Thus, these values will overestimate the frequency with which potentially targetable pathogenic mutations will be found in these kinases.

We also considered the genes defined to be members of the RAS pathway by the NCI RAS Initiative[40]. The Ras pathway has a clear role in human cancer, with mutations in this pathway capable of conferring multiple hallmarks of cancer[48,49]. Additionally, the scientific literature commonly estimates 30–33% of all cancers have a mutation in either *KRAS*, *NRAS*, or *HRAS*[50–52]. In contrast, our integrated genomic-epidemiology approach provides a revised estimate of 15% of cancers harboring a *KRAS*, *NRAS,* or *HRAS* mutation – less than half of those previously reported estimates. Our findings on the proportion of all cancers

with a RAS gene mutation are consistent with a recent analysis that performed a simpler weighted analysis of *KRAS*, *NRAS*, and *HRAS* mutations. This study, which also included copy number variants, estimated that 19% of all cancer patients have a mutation in one of these *RAS* genes[53].

Ongoing cancer driver gene discovery efforts have identified many new drivers within the RAS pathway[27,54,55]. Like our analysis of previously designated cancer drivers and protein kinases, our analysis of RAS pathway gene mutations finds a handful of commonly mutated genes followed by a long-tail of less frequently mutated genes (Fig. 3C; Supplementary Data 3). Thus, although it is possible that more genes in the RAS pathway will be classified as cancer driver genes, it does not appear that

**Table 1 Overall mutation rates for consensus cancer genes.**

| Gene | Mutation proportion (%) | The most common cancers to harbor a mutation in this gene (ROSETTA classification, % of all instances of this mutation) |
|---|---|---|
| *TP53* | 34.5 | Colorectal Adenocarcinoma (16.0%), Breast Carcinoma (15.2%), Lung Adenocarcinoma (11.6%) |
| *PIK3CA* | 13.5 | Breast Carcinoma (38.0%), Colorectal Adenocarcinoma (19.2%), Endometrial Cancer (10.6%) |
| *LRP1B* | 13.1 | Lung Adenocarcinoma (21.7%), Malignant Melanoma (17.1%), Colorectal Adenocarcinoma (13.1%) |
| *KRAS* | 10.5 | Colorectal Adenocarcinoma (35.0%), Lung Adenocarcinoma (23.1%), Pancreatic Adenocarcinoma (22.7%) |
| *APC* | 10.1 | Colorectal Adenocarcinoma (65.7%), Malignant Melanoma (6.0%), Prostate Cancer (5.1%) |
| *FAT4* | 9.5 | Colorectal Adenocarcinoma (22.6%), Malignant Melanoma (19.3%), Lung Adenocarcinoma (12.9%) |
| *KMT2D* | 9.2 | Colorectal Adenocarcinoma (13.9%), Urothelial Cancer (12.8%), Prostate Cancer (10.1%) |
| *KMT2C* | 9.1 | Breast Carcinoma (14.3%), Colorectal Adenocarcinoma (13.9%), Lung Adenocarcinoma (12.5%) |
| *BRAF* | 7.6 | Malignant Melanoma (35.3%), Colorectal Adenocarcinoma (21.7%), Thyroid Carcinoma (20.6%) |
| *ARID1A* | 7 | Endometrial Cancer (18.0%), Colorectal Adenocarcinoma (15.5%), Urothelial Cancer (15.0%) |
| *FAT1* | 6.6 | Colorectal Adenocarcinoma (16.2%), Lung Adenocarcinoma (14.2%), Malignant Melanoma (10.2%) |
| *PTEN* | 6 | Endometrial Cancer (30.3%), Breast Carcinoma (13.7%), Colorectal Adenocarcinoma (12.0%) |
| *ATM* | 5.5 | Colorectal Adenocarcinoma (21.0%), Lung Adenocarcinoma (12.7%), Prostate Cancer (10.0%) |
| *ZFHX3* | 5.4 | Colorectal Adenocarcinoma (21.4%), Malignant Melanoma (12.6%), Endometrial Cancer (12.5%) |
| *CREBBP* | 5.3 | Follicular lymphoma (15.7%), Colorectal Adenocarcinoma (14.7%), Urothelial Cancer (11.2%) |
| *GRIN2A* | 5.1 | Malignant Melanoma (26.8%), Lung Adenocarcinoma (14.8%), Colorectal Adenocarcinoma (14.3%) |
| *NF1* | 5 | Lung Adenocarcinoma (16.4%), Malignant Melanoma (15.4%), Colorectal Adenocarcinoma (13.2%) |
| *TRRAP* | 4.9 | Colorectal Adenocarcinoma (22.0%), Malignant Melanoma (18.0%), Lung Adenocarcinoma (11.1%) |
| *PDE4DIP* | 4.9 | Colorectal Adenocarcinoma (19.3%), Malignant Melanoma (15.2%), Lung Adenocarcinoma (11.2%) |
| *PTPRT* | 4.9 | Malignant Melanoma (26.5%), Colorectal Adenocarcinoma (18.7%), Lung Adenocarcinoma (16.2%) |
| *RNF213* | 4.8 | Colorectal Adenocarcinoma (25.0%), Malignant Melanoma (12.8%), Lung Adenocarcinoma (8.8%) |
| *PREX2* | 4.7 | Malignant Melanoma (24.5%), Colorectal Adenocarcinoma (16.7%), Lung Adenocarcinoma (10.2%) |
| *SPEN* | 4.5 | Colorectal Adenocarcinoma (16.6%), Malignant Melanoma (14.0%), Breast Carcinoma (12.1%) |
| *KMT2A* | 4.4 | Malignant Melanoma (18.3%), Colorectal Adenocarcinoma (17.6%), Urothelial Cancer (11.8%) |
| *ERBB4* | 4.4 | Malignant Melanoma (18.9%), Colorectal Adenocarcinoma (17.9%), Lung Adenocarcinoma (15.7%) |

The twenty-five genes from the Cancer Gene Census, Tier 1, that are most commonly mutated in cancer for the US population. In addition to the mutation proportion, the three types of cancer that contribute the most to individuals with that gene mutated are listed. For each of the three, the proportion of all estimated incidences of that gene that are from the listed ROSETTA histology is indicated parenthetically.

there are pan-cancer high prevalence RAS pathway drivers remaining to be discovered. Altogether, our analyses suggest that cancer genomics is less dominated by common, high-frequency, driver gene mutations than studies of individual cancers have implied.

**Pan-adenocarcinoma and pan-squamous cell carcinoma analysis.** Cancers can be grouped into higher-level groupings on the basis of shared histology, such as adenocarcinomas, lymphomas, and gliomas. Within ICD-O-3, the different histology codes are organized within such higher-order categories[36]. Our ROSETTA implementation pragmatically employed varying levels of granularity within the different higher-order categories as part of our effort to make accurate population-level estimates on the basis of the available sequencing data. We next analyzed the sequencing data at the level of these higher-level groupings to evaluate how the patterns of gene mutation proportions varied between the more general cancer classifications.

The two most common higher-level classes of cancer are adenocarcinomas and squamous cell carcinomas (SCC) (Fig. 4a). We calculated pan-adenocarcinoma and pan-SCC mutation proportions by performing our weighted epidemiological analysis, but limited to all ROSETTA adenocarcinoma (Fig. 4b, Supplementary Data 4) and all ROSETTA SCC categories, respectively (Fig. 4c, Supplementary Data 4). We considered the epidemiologically weighted proportions of COSMIC Cancer Gene Census (Tier 1) mutations across all types of adenocarcinoma (Fig. 4d) and across all forms of malignant squamous cell carcinomas (Fig. 4e). We also directly compared rates of mutations between each of these two major subtypes of carcinoma (Fig. 4f). For most genes, the mutation frequency within adenocarcinomas and within squamous cell carcinomas were within 5% of each other. The only two genes that had a net rate of mutation that was more

than 5% higher in adenocarcinoma were *KRAS* and *APC*. *KRAS* is estimated to be mutated in 14% of adenocarcinomas, but only 1% of SCC, and *APC* is calculated to be mutated in 13% of adenocarcinomas and 5% of SCC. Seven genes favored SCC with overall mutation rates that were 5% higher than their rate in adenocarcinoma. *TP53* is estimated to be mutated in 63% of all SCC and 34% of all adenocarcinomas, which makes *TP53* the most frequently mutated Consensus Cancer Gene in each of the two major classifications of cancer. The other genes that were much more mutated in SCC (with estimated mutation proportion in SCC and in adenocarcinoma, respectively) were *LRP1B* (23%, 10%), *KMT2D* (16%, 7%), *FAT1* (15%, 5%), *CDKN2A* (12%, 2%), *NOTCH1* (10%, 3%), and *NFE2L2* (10% and 1%).

The next two major groupings of cancer within our ROSETTA subclassification of ICD-O-3 categories are melanomas and transitional cell carcinomas (Supplementary Data 4). We evaluated mutation proportions for each of these next most abundant subclassifications (Fig. 4g, h). The most frequently mutated gene in melanoma was *BRAF*, which made melanoma the only major category of cancer for which *TP53* was not the most commonly mutated gene and for which *PIK3CA* was not the most common oncogene. Additionally, transitional cell carcinoma was notable in that seven of its ten most commonly mutated genes have direct roles in epigenetic and transcription factor regulation.

We also considered the mutation proportions of the three *RAS* genes (*KRAS*, *NRAS*, and *HRAS*) within these four general classifications of solid tumors. We found that each was the most mutated of the three in at least one major type of cancer (Fig. 4i). *KRAS* was the most frequently mutated of the three in adenocarcinoma, and *NRAS* was the most frequently mutated in melanoma. Interestingly, *HRAS* was the most commonly mutated *RAS* GTPase in both squamous cell carcinoma and

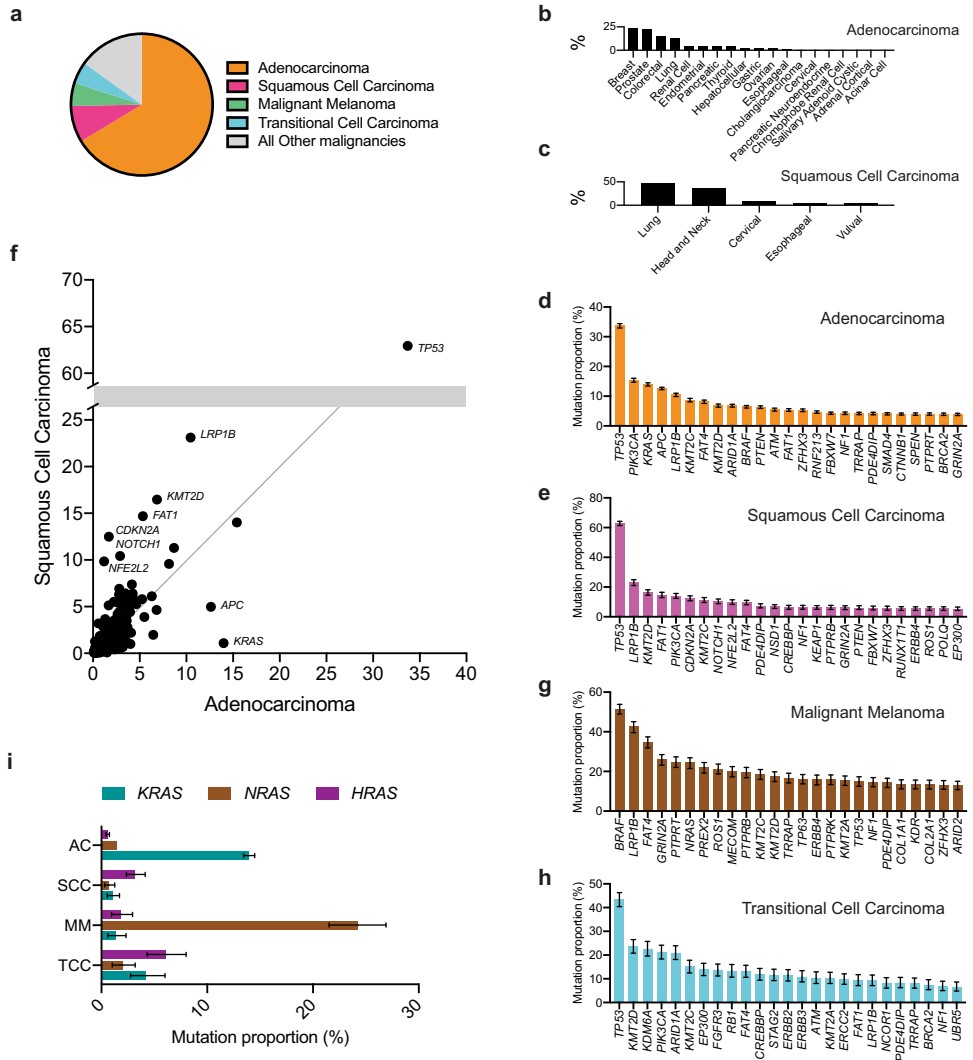

**Fig. 4 Analysis of gene mutation proportions within common subclasses of cancer. a** Proportion of all cancers in epidemiological studies that are within the major sub-classifications of adenocarcinoma (AC), squamous cell carcinoma (SCC), malignant melanoma (MM), or transitional cell carcinoma (TCC). **b** Specific types of AC, with their relative proportion of all ACs. **c** Specific types of malignant SCC, with their relative proportion of all SCCs. **d** Top 25 mutated genes in epidemiologically weighted, pan-AC analysis. **e** Top 25 mutated genes in epidemiologically weighted, pan-SCC analysis. **f** Comparison of mutation proportions between AC and SCC for the Cancer Gene Census genes. Selected genes with non-overlapping 95% confidence intervals are indicated. All genes and their mutation proportions with confidence intervals are listed in Supplementary Data 4. **g** Top 25 mutated genes in epidemiologically weighted pan-melanoma analysis. **h** Top 25 mutated genes in TCC. **l** Proportion of each of the major cancer subtypes that have *KRAS*, *NRAS*, and *HRAS* mutated. Error bars in all panels **d**, **e**, and **g–i** represent 95% confidence intervals determined through simulated samples ($n = 2,000$ independent Poisson distributed computational samples for each gene and histology with the calculated mutation proportion as the central value), and bar levels represent calculated mutation proportions.

transitional cell carcinoma. *KRAS* is commonly considered the worst of the three *RAS* GTPases for cancer promotion; our pan-cancer analysis rather suggests that the issue of which oncogenic *RAS* GTPase is most cancer-promoting may be dependent upon the type of cell in which cancer develops.

## Discussion

Overall, this work describes our approach to integrating epidemiological and genomic data to provide real-world estimates for the overall proportion of US cancer patients who harbor a coding exonic mutation in each gene. We focused here on data for cancer incidence, defined as new cases within a period of time. Epidemiological data is also available for prevalence (the number of individuals alive at a given time who have a disease) and mortality (the number of cancer deaths within a period of time). We chose

not to calculate epidemiologically weighted estimates for mortality and prevalence because clinical outcomes can depend upon the specific genetic mutations present in the cancer[56–58]. Here, we assumed that the likelihood of being sequenced is more a function of incidence than of survival. However, the decision of which tumors to sequence is likely influenced by other factors, such as tumor size and staging. Those choices and practices, which are beyond the scope of the present work, could introduce biases into the underlying data that are then reflected in the calculated mutation proportion estimates. With the ROSETTA framework now established, it should be possible to adapt this epidemiological weighting approach to other types of sequencing data to investigate copy number alterations, intronic mutations[39], gene fusions[59], and methylation patterns[60].

We highlight that the gene mutation proportions presented here, though calculated from high-quality epidemiological and

genomic data, should still be considered estimates. Cancer genomic studies may not necessarily be accurate representations of all individuals within a population who have a specific type of cancer, whether due to racial, ethnic, and/or geographical differences[61], exposure to carcinogens, and age of onset[62]. Additionally, the finer histological subdivisions of a major class of cancer may not be proportionally represented within the entire collection of tumor samples for a given form of cancer. Our ROSETTA mapping is not the only conceivable approach for integrating epidemiological and genomic data, and others could employ the same general strategy used here but make different choices. We anticipate that estimates will continue to improve as cancer genomics becomes increasingly common, as sequenced samples are increasingly better annotated, and as rare cancers are increasingly sequenced. At some point, tumor sequencing may be so common that sampling is not even necessary, as all cancers within a population would receive sequencing upon initial detection. That situation would overcome the need to extrapolate from a sample to calculate an estimate. It would overcome the need to make assumptions about the genetics of cancer types that have not been sequenced. Additionally, that situation would also overcome possible biases in the data from how and when tumors are selected to be a representative sample for a form of cancer and sent to sequencing. Until all patients have their tumors sequenced, we believe that the values presented here are useful estimates for cancer researchers who desire to know how commonly their gene(s) of interest is (are) mutated in cancer.

*RAS* mutations have commonly been stated to be found in 30–33% of cancers[50–52]. We have shown that *RAS* mutations are rather found in ~ 15% of all cancers, supporting trends observed in previous, non-epidemiological weighted pan-cancer analyses[26,27,33], and consistent with a recent, simpler, epidemiologically weighted *RAS* mutation estimate[53]. The discrepancy between historical estimates and the more recent estimates is reasonable when one considers that the two most common cancers, breast, and prostate carcinoma, which together comprise 29% of all new cancer diagnoses in the USA, rarely have *RAS* gene mutations. Additionally, the relatively common cancer with the highest prevalence of *RAS* mutations in pancreatic adenocarcinoma, which has a KRAS mutation in nearly 90% of all sequenced samples. However, the impact of pancreatic adenocarcinoma on the population-level estimate is limited because < 3% of new cancer diagnoses in a given year are for pancreatic adenocarcinoma[34,35]. Therefore, it appears that the commonly encountered estimates for *RAS* mutation prevalence failed to account for the fact that the cancers that commonly harbor *RAS* mutations are not as common as the cancers that rarely have *RAS* mutations. The importance of obtaining accurate estimates for *RAS* genes, as well as for other genes, can be inferred from two recent, high-profile, high cost research efforts: the NCI RAS Initiative[40] and the DARPA Big Mechanism program[52,63]. Both were in part justified by the high abundance of RAS mutations relative to other oncogenes. Thus, our presentation of accurate epidemiological estimates may influence future efforts that aim to focus on the common mutant drivers of cancer.

The lower than expected rate of *KRAS* mutations also highlights that there are few high-frequency mutations in human cancer when one looks across all cancer cases. This has implications for drug development, where small molecular targeted therapies that target common oncogenic mutations have become a major focus. Although these drugs clearly offer benefits to many individuals, this analysis suggests that the proportion of all cancer patients who will benefit from any one targeted therapy designed to target a specific mutated gene product will be limited. The surprising dearth of high-frequency pan-cancer drivers also has important implications for cancer development, as it suggests cancer may be less dominated by high-frequency drivers than

studies of specific cancers have implied. This further suggests that other factors not considered here, such as copy number variation, epigenetic dysregulation other than by mutation of epigenetic genes, fusion proteins, and microenvironmental cues, may all play large roles in the convergent cancer phenotype[48,49]. Approaches that identify the convergent phenotypes and their targetable vulnerabilities may offer a better chance to benefit many patients than efforts that focus only on specific mutations[64,65].

## Methods

**Cancer Histological Reclassification.** We developed a mapping between pairs of ICD-O-3 morphology (histology) and topography (anatomical location) codes and between the tumor classification descriptors used in cancer genomics that we termed ROSETTA (Reclassification Of Sequencing and Epidemiological Tumor Type Annotations). To do this, two medical doctors with training in pathology, cancer biology, and bioinformatics, systematically and iteratively went through ICD-O-3 and cancer genomic annotation files to assess what level(s) of cancer groupings would be needed to resolve the data sets. ROSETTA categories were developed to serve as a common cancer classification system onto which both epidemiologic and genomic cancer classifiers could be mapped. We then specified maps from ICD-O-3 codes to ROSETTA (Supplementary Document 1) on the basis of our reclassification. Additionally, we developed maps from cancer patient files to ROSETTA on the basis of patient metadata. ROSETTA categories were developed to remain consistent with major organizational principles of ICD-O-3.

**Epidemiological Data and Processing.** Cancer epidemiological data were obtained from the NCI Surveillance Research Program database SEER using SEER*Stat software. We utilized the database named: Incidence - SEER Research Data, 18 Registries, Nov 2019 Sub (2000-2017). We exported data for malignant diagnoses in table output with rows specified by ICD-O-3 Histo/behave codes and columns specified by Site recode ICD-O-3/WHO 2008 location codes.

There were multiple steps to processing (Supplementary Document 1). Briefly, steps included grouping into higher level anatomical location site codes using the level of resolution employed by the ACS in their annual cancer epidemiology report[35], processing samples with unclear histology and unclear site of origin, and reclassification by ROSETTA. After each processing step, we would check that the total number of samples within the study was constant to ensure no loss of data in our implementation. The output of our processed SEER data was a table where the rows were 43 sites (anatomical locations) and the columns were 370 ROSETTA classifications. The values in the table were the estimated counts for each histology at that location.

**Cancer Genomic Data.** We utilized public, non-embargoed, and non-provisional cancer genomics studies that were included on cBioPortal[29,30]. We included exome sequencing studies on tissue samples, i.e., not from xenografts or cell lines (Supplementary Data 2). We focused on nonsense and missense mutations, as well as small indels. Within the cBioPortal notation, we counted a gene within a patient as mutated if it had one (or more) of the following mutations called: nonsense_mutation, frame_shift_del, frame_shift, frame_shift_ins, missense_mutation, missense, nonsense, in_frame_del, in_frame_ins, nonstop_mutation. We did not include splice site or fusion mutations. We re-annotated Mutation Annotation Format (MAF) files for each patient sample with the appropriate ROSETTA annotation. The output of our genomic processing step was a table that listed the total number of samples with a mutation in each gene, done for every ROSETTA cancer classification. When a study included longitudinal samples (i.e., where the same patient was sampled several times) we utilized only the first sample. When multiple studies included data from the same patient sample, we utilized the mutation data from the most recent study. As the same gene may be referred to with different names, we also standardized gene names using a human gene names map described in the cBioPortal documentation. Cancer genomic data utilized come from a variety of sources, including the TCGA and TARGET programs. The results published here are in part based upon data generated by the TCGA Research Network: https://www.cancer.gov/tcga, and by the Therapeutically Applicable Research to Generate Effective Treatments (https://ocg.cancer.gov/programs/target) initiative, phs000218: https://portal.gdc.cancer.gov/projects. A list of all genomics studies utilized is provided within a table in Supplementary Data 2. That table also lists which ROSETTA code(s) were assigned to one or more samples within each of the genomics studies. Text files that map from each sample to a ROSETTA code are available in the supplementary software. Additional information on mapping and software implementation is provided in the "Supplementary Methods".

**Mutation Proportion Estimates.** We converted our genomic output table to a table that listed the observed proportion of sequenced tumors within a ROSETTA classification that harbor a given gene mutation. In other words, we converted our genomic output table into an $m \times n$ matrix (**C**), where $m$ is the number of genes and $n$ is the number of ROSETTA classifications that have been sequenced. In our

current study, the value of $m$ is 21,271 and the value of $n$ is 73. The values in **C** represent the conditional probability that a gene is mutated given the ROSETTA classification.

We converted our SEER epidemiological output to list the number of all representative cancers from each ROSETTA code. This resulted in a $k \times 1$ matrix, **S**, where $k$ is the number of all ROSETTA classifications (370) and $k > n$ as not all ROSETTA classifications have representative sequencing data. We had representative sequencing data for ROSETTA codes that account for 93% of all observed human cancers. We assume that the weighted proportions of mutations for the 93% of all cancers are a good estimate for the weighted proportion in all cancers. We thus converted **S** to **S'**, an $n \times 1$ matrix by eliminating all rows that were for a ROSETTA classification where there were no representative genomic sequencing data. We normalized the vector, so it summed to 100%.

The estimated mutation proportion could then be calculated by summing the product of the mutation rate in a ROSETTA classification by the proportion of all cancers that have that ROSETTA classification for all ROSETTA classifications. Alternatively, **C•S = G**, where **G** is an $m \times 1$ matrix that lists the overall, weighted mutation frequencies of all genes (i.e., Fig. 1, bottom).

We performed statistical analyses with the genomic dataset by constructing in silico studies of cancer mutation observations. Since typical genomic studies (total 19,181 samples) were two or more orders of magnitude smaller than the epidemiological studies (total 7,167,808 samples) in any given tissue or histology, we take the genomic studies to be the primary source of any variations in our results. We performed statistical analysis with the assumption that Poisson distributions applied to the cancer cases observed over the study periods. We generate two thousand Poisson distributed samples for each combination of gene and histology with central value as calculated from genomic studies included. Replicates for the proportion of mutated cases for each gene are calculated by processing each in silico sample through our reweighting (conditional probability) pipeline (Fig. S2), and 95% confidence intervals were calculated from these replicates.

**Reporting Summary**. Further information on research design is available in the Nature Research Reporting Summary linked to this article.

## Data availability
No new data were generated in this study. The input genomic data is publicly available from www.cbioportal.org, and URLs for all included genomic studies are listed as a table in the document, supplementary data 2. The input epidemiological data is publicly available for download via the SEER-Stat program that is available upon registration with NCI SEER via the website https://seer.cancer.gov/seerstat/. All needed genomic data to reproduce this work is available as ROSETTA relabeled (processed) genomic data that is present in the zipped file Genomics_Analysis\interim_files\df_mutfiles.zip in Supplementary Software 1. All epidemiological data needed to reproduce this work is available as ROSETTA relabeled (processed) epidemiological data that is present in file SEER_Analysis/Output_SEER.txt in Supplementary Software 1. All output data described in the manuscript is provided in the "Supplementary Information/Supplementary data".

## Code availability
The source code used to combine genomics and epidemiological data and generate all of the figures and tables in the manuscript is present as a supplementary zipped folder and can also be downloaded from the link, https://github.com/GMendiratta/ROSETTA-for-Cancer-Mutations. The source code to download and process genomic data and the source code to process raw SEER data are also included in folders named Genomics_Analysis/ and SEER_Analysis, respectively. The code is written in python 3 using Jupyter notebook IDE and uses NumPy, pandas, random, and Matplotlib libraries. This code is provided as-is and may be copied, re-used, edited with a citation to this manuscript. No additional permission from the authors is necessary.

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

## Acknowledgements

The authors thank Hannah Carter, Olivier Harismendy, Jason Sicklick, Pablo Tamayo, Peter Salamon, and the members of the Stites Laboratory for helpful conversations and comments. This work was supported by Pioneer Fund Postdoctoral Scholar Award, NIH K22CA216318, NIH T32CA009370, NIH P30CA014195, and the Melanoma Research Alliance Young Investigator Award.

## Author contributions

E.K., M.A., G.M., and E.C.S. designed the project. M.A. and E.C.S. developed the ROSETTA mapping between epidemiological and cancer sequencing data to a shared alternative cancer classification. M.A. and G.M. developed the computational processes for integrating ROSETTA into epidemiological and genomic data sets. E.K., D.L., and G.M. aggregated and processed cancer genomic data. G.M., M.A., and E.C.S. analyzed the epidemiological data. G.M. integrated the epidemiological data with genomic data. G.M. and E.C.S. performed statistical analyses. D.L. and M.T. assisted with testing code, validation, and documentation. E.C.S. and G.M. wrote the manuscript with input from the other authors.

## Competing interests

The authors declare no competing interests.
