## [Peer Review File · Nature Communications]

REVIEWER COMMENTS

Reviewer #1 (Remarks to the Author): Expert in epidemiology and statistics

There are two areas where the authors might consider strengthening the manuscript.

The first is to provide compelling evidence that better mutation rate estimates across all cancers are important for public health and oncology drug development. In particular, although there are basket trials for the determination of whether a drug is active (i.e. for early phase trials), the subsequent phase III trials are most often done within a specific tumor type. Most often because targeting a mutation does not have the same effect across all tumor types, which is ascertained during the early phase basket trial. Hence, it is hard to see how having more accurate estimates of mutation rates would change this. It would be useful if the authors could provide an example where a more accurate mutation rate has led to informative results or a change in drug development. In addition, it likely does not matter whether the rate is improved by a small amount but what matters is whether the mutation is known and whether it is targetable. There have been several targeted therapies that only apply to a very small subgroup within a specific cancer (so the overall mutation rate would be low). More literature support is needed to make it more convincing that having a more accurate cancer mutation rate across ALL cancers is really a significant advance in the field.

It is true that the proportion of different tumor types that are sequenced do not reflect the incidence of the specific cancers and to get a better estimate of the mutation rate across all cancers, there needs to be an adjustment to reflect this. However, there are several concerns with this approach and assumption that this is necessary.

(*) It does not appear as though the proposed reclassification scheme used for sequenced tumors as well as SEER data has been validated. It would strengthen the manuscript if there is evidence that the scheme that is used results in accurate classifications of tumors (from both SEER and from sequencing studies).

(*) The SEER data reports incident cancers whereas the sequencing data are not necessarily from incident cancers. It is not clear that adjusting rates obtained from both incident and prevalent cancers by rates of yields an accurate estimate.

(*) It is known that cancers that are sequenced differ from those that are not in meaningful ways and these differences may impact the mutation rates. Hence, merely adjusting to reflect the proportion of incident cancers may not yield accurate estimates of mutation rates across all cancers. One such difference is that sequenced tumors tend to be larger than those not sequenced and so they tend to reflect more advanced disease. There is evidence that mutation rates are different in advanced disease versus early disease. The proposed calculation cannot adjust for this. Hence, it is not clear that it truly is an accurate estimate. In addition, some of the sequenced tumors may have already been treated and the mutations may reflect mechanisms of drug resistance.

(*) Another concern is whether it truly matter what mutation rates across ALL cancers. For example, in SEER, most of the breast cancer cases are early stage due to screening, which captures cancers at an early time point. A significant portion of these cancers are cured with surgery and available treatments. Hence, does it matter what the mutation rate is in this cancer (at least from a treatment perspective)? This assumes that the mutation rates are the same in these cancers as in more advanced disease and in cancers for which there are no treatment options. Raises the question of whether pooling across all cancers really does make sense as well as whether it is accurate to pool across all cancers (when even within different subtypes of cancers, there are differing mutation rates, which is not captured in this calculation).

(*) Assuming that non-represented cancers have the same mutation rates as the other cancers is questionable. It is known that tumor burden (as measured by the number of mutations within a tumor) differs by cancer.

Reviewer #2 (Remarks to the Author): Expert in cancer genomics and bioinformatics

This is an elegant study tackling an important unresolved question: how do the frequency of somatic driver mutations reported by large pan-cancer consortia compare to that in the broad population of cancers. To address it, the authors developed a mapping strategy they call ROSETTA, then apply it to two versions of TCGA pan-cancer data. The study is generally well-written, and admirably documented with robust and very valuable supplementary.

1. Terminology

An absolutely critical error throughout is the use of the term "mutation". The authors appear to only be analyzing one of the several classes of mutations -- "somatic functional single nucleotide variants (SNVs)". Almost every conclusion in this paper is incorrect when phrased generally as mutation because of the large number of translocations, inversions, gains, deletions and so forth. It is fine in colloquial terminology to speak of "driver mutations" when only referencing SNVs, but in the literature this would be a critical error. The entire paper absolutely needs to be updated to use the technically correct terminology, including a title change.

2. Gene Symbols

Supplementary tables show gene symbols like gene symbols of 1-Dec and 3-Mar that appear to be artifacts of Excel processing

3. ROSETTA

Significantly more information is required on ROSETTA:

a) How was it validated? Please attach estimates to the error rate of the mapping process.

- b) Where is there a list of all TCGA samples with their original IDs and new ROSETTA classifications?
- c) Please provide a software implementation or equivalent that could be used to map datasets beyond TCGA (e.g. PCAWG or GENIE)

4. Visualization

Please ensure scatterplot axes are exactly matched in their range and tick-marks, and include $y=x$ lines to allow direct comparison

5. Statistics

- a) It is unclear in cases like the adenocarcinoma vs. squamous cell carcinoma if the inferred differences are actually statistically significant. One presumes so, but it is difficult to ascertain.
- b) Similarly please attach confidence intervals to population driver estimates based on the accuracy of estimating driver frequencies from a few hundred cases per cancer type.

Reviewer #3 (Remarks to the Author): Expert in epidemiology and statistics, reviewed with Reviewer #4

The authors generate mutation rates among all cancers within the US population. They note that in prior works with COSMIC, TCGA, GENIE, the representation of difference cancers in the resources is not provided and is not designed to be a proportional relative to those cancers in a population. The authors develop a technique they call ROSETTA (Reclassification Of Sequencing and Epidemiological Tumor Type Annotations), which appears to interconvert between the different nomenclatures used to label cancer diagnoses in the SEER database and several genomic studies from cBioPortal. The authors show that certain genes are commonly mutated (eg p53); but there is a long tail, and there are many other mutations that may be present.

Overall, this is interesting work. The writing is generally clear. There are almost no typos.

The authors should be commended for trying to do this large-scale analysis, and I hope my comments are helpful.

Major comments:

(1) I think the methods need to be clarified, with more details of each step, and more description in the text and figures. I reviewed this manuscript in detail with a computational biostatistician who has knowledge of the genomic databases, to complement my own understanding of SEER, and we think the authors are doing the following, but we need more clarification. I'm going to summarize the steps we think the authors are using, and provide questions within each step:

If I understand correctly, in step 1 entries in the table are observed counts based on SEER data. How did the authors come up with the groupings in Sup Document 1?

How are histologies besides those listed included in the current work? It appears, based on Sup File 1 and Figure 4 that most of the histologies are distilled to adeno vs SCC vs melanoma vs TCC. However, is there a reason this figure doesn't include small round blue cell tumors, gliomas, lymphomas?

I wonder if the authors could better summarize Sup File 1 to provide a frequency table of the codes, and then provide a figure showing how the histology codes are grouped and the rationale for grouping.

In step 2, it looks like they use the ROSETTA class scheme to create matrix vs the anatomical location. The entries are estimated counts based on entries from Step 1. My comments for this step are:

Why use "site" in step 1, and use "anatomical location" in step 2?

How do the authors account for multiple entries within each column or row? eg, if within lung cancer, there is non small cell lung cancer, adeno, squamous cell, small cell, how are these subdivided?

In Step 3, they bring in mutation data, creating a matrix of genes vs ROSETTA classes. This seems clear, but I think I need to understand the prior steps better.

In Step 4, the ROSETTA classes and genes are converted to percentages. Entries are column frequencies based on counts of the matrix in Step 3. This step depends on having mutation data and SEER data. For certain cancers, it sounds like it is possible that there are values of 0, if there were no genomic data for that cancer. How do the authors account for this? How often does it occur?

In Step 5, in each ROSETTA class, they look at the total number of SEER samples, based on Step 2. There might be ROSETTA data for cancers we don't have genetic data, producing a value of 0. The data are scaled proportionately. The n are converted to percentages.

Step 6. Take the genes (in rows) vs classes of ROSETTA (in columns), and multiply this by ROSETTA classes, to provide a weighted gene mutation frequency.

Is this understanding correct?

(2) Comments about tables and figures:

Table 1 is too difficult to understand with so much text. Could this be transformed to be a figure?

Figure 1: how are patients with multiple mutations included in this work?

Figure 3: it looks like there is a long tail of mutations, and mutation frequency of any gene is ~3%. Could it be that these mutations are normal and expected? Or are they all expected to cause pathology?

I am having a difficult time understanding supplementary file 1

(3) Minor comments:

Carcinoma misspelled in Figure 4D

Reviewer #4 (Remarks to the Author): Expert in epidemiology and statistics, reviewed with Reviewer #3

Key Results

The authors have developed ROSETTA, a novel method for classifying tumor types that enables them to match epidemiologic data from SEER with cancer genomic data from the CBioPortal. Using ROSETTA, they compute population-based estimates of gene mutation frequencies across all cancers in the U.S. as well as specific types of cancer, e.g., all squamous cell carcinomas. These population-based mutation frequencies are then compared to gene mutation frequencies obtained in two TCGA “PanCan” studies as well as other previous publications. The findings suggest that in the general population, mutations in cancer affect a few driver genes with high frequency as well as a large number of genes with low frequency (long tail). Specific findings include a high frequency of TP53 mutations, a lower than expected frequency of KRAS mutations, and common mutations of epigenetic modifiers KMT2C, KMT2D, and ARID1A.

Validity

I have no concerns.

Significance

As the authors note, ROSETTA represents one possible classification scheme that could be used to merge epidemiologic and genomic data sets, and because of this the mutation frequencies presented in the paper should not be viewed as definitive. Regardless, ROSETTA is a principled and well-developed method, and the fact that the authors have applied it to obtain population-level gene mutation frequencies using data from SEER and high-profile cancer genomics studies is noteworthy and likely novel.

Data and Methodology

The authors utilize publicly available data from SEER and the CBioPortal, both of which are widely used. Thus there are no concerns about data quality. The approach detailed in the Methods section produces the desired weighted gene mutation frequencies.

Analytical Approach

See the above comments regarding Methodology.

Suggested Improvements

- While it is natural to reference the two TCGA PanCan papers, the authors use the terms “not well estimated” and “large errors” when comparing their population-based mutation frequencies with the mutation frequencies in these studies. This language suggests that the mutation frequencies from the PanCan studies should be directly comparable to the population-based mutation frequencies computed by the authors, which is not the case. Please clarify.
- The authors note that the high frequency of TTN and OBSCN mutations may arise because of the large sizes of these genes. With that in mind, it would be useful to know whether these mutations are often pathogenic or simply passenger mutations.
- Related to the previous comment, it would be interesting to know how much the population gene mutation frequencies change if you restrict to pathogenic mutations. Perhaps this can be computed readily after filtering the MAFs.
- The lower than expected frequency of KRAS mutations is noted in the paper. Can you provide any insight into why this occurs? Presumably there are specific tumor types that are common in the population yet have low RAS mutation frequencies, but I don't think this is ever explicitly stated.
- The phrase “Cancer sequencing data, once analyzed by a variety of statistical approaches” suggests that statistically-based methods are no longer applied when analyzing sequencing data. This is not the case, as evidenced by the widespread usage of MutSig and other related methods. Please clarify this point.
- The authors note that genes with mutation frequencies above 8% were either on the Consensus Cancer Gene Census list or were very large proteins. How was the 8% cutoff chosen? Also, only a handful of genes satisfy this criteria, so perhaps they should be listed in the text.
- It looks like there is considerable overlap between the percentages in Table I and the percentages shown in Figure 3A and Figure 4D. Perhaps this table could be moved to the Supplement.
- In the Methods section the authors refer to the $m \times n$ matrix C and the $k \times 1$ matrix S . Why not specify the values of m , n , and k ?
- Not all readers will be familiar with matrix multiplication or how it is being used in the current context to compute population-level mutation frequencies. Because of the large number of ROSETTA classifications, I recognize that it may not be feasible to show a real worked example. However, even a toy example would be useful. Perhaps this could be added to the supplement.
- The discussion of gene mutation frequencies in melanoma and transitional cell carcinoma seems brief in comparison to the discussion of mutation frequencies in adenocarcinoma and squamous cell carcinoma. For example, the authors note that TP53 is not the most commonly mutated gene and PIK3CA is not the most commonly mutated oncogene in melanoma without actually naming the most commonly mutated gene or most commonly mutated oncogene. Admittedly this information is presented in Figure 4, but it should appear in the text as well.

Clarity and Context

I have no concerns.

References

I have no concerns.

My Expertise

Although I have worked with SEER data in the past, my role was primarily that of a data analyst. As such, I am not well-versed in the cancer epidemiology literature, and thus I cannot assess whether similar studies have been performed and should be referenced.

REVIEWER COMMENTS

Reviewer #1 (Remarks to the Author): Expert in epidemiology and statistics

There are two areas where the authors might consider strengthening the manuscript.

We thank Reviewer #1 for the helpful review and the suggestions for ways to strengthen the manuscript. The updates to the manuscript (detailed below) should make it read better, more clearly communicate the value of the approach, more clearly communicate the methods and validation, and more clearly communicate assumptions and possible biases in the underlying data and methods.

The first is to provide compelling evidence that better mutation rate estimates across all cancers are important for public health and oncology drug development. In particular, although there are basket trials for the determination of whether a drug is active (i.e. for early phase trials), the subsequent phase III trials are most often done within a specific tumor type. Most often because targeting a mutation does not have the same effect across all tumor types, which is ascertained during the early phase basket trial. Hence, it is hard to see how having more accurate estimates of mutation rates would change this. It would be useful if the authors could provide an example where a more accurate mutation rate has led to informative results or a change in drug development.

We thank Reviewer #1 for the thoughtful and helpful review. Within the revised manuscript, we have substantially increased our explanation for why this information is important. Rather than have these reasons as a small part of a bigger paragraph (as in our original submission), we have now expanded the arguments into two stand-alone paragraphs. We have used the additional space to elaborate and provide additional examples, details, and references. With regards to basket trials, we highlight the FDA approvals of NTRK inhibitors and pembrolizumab, which were done in a cancer histology agnostic manner with substantial support coming from basket trials. We have included references on basket trials and these FDA approvals and motivating trials.

We have also elaborated on the potential public health benefits of this information. For example, we highlight how the NCI RAS Initiative and DARPA RAS “Big Mechanism” program were both heavily motivated by the idea that RAS was mutated in 30% of all cancers. We also now highlight that personalized medicine efforts that use agents off-label could benefit from knowing pan-cancer frequencies so they can better estimate how often they may encounter certain genes mutated.

In addition, it likely does not matter whether the rate is improved by a small amount but what matters is whether the mutation is known and whether it is targetable. There have been several targeted therapies that only apply to a very small subgroup within a specific cancer (so the overall mutation rate would be low). More literature support is needed to make it more convincing that having a more accurate cancer

mutation rate across ALL cancers is really a significant advance in the field.

We agree that not all variants to a gene will be drivers or targetable. In our revised manuscript, the interpretations of our findings better highlight that our numbers are upper bound estimates for driver and targetable variants. We also highlight that the low epidemiologically weighted, pan-cancer values may suggest a more difficult road forward for personalized medicine than other studies have implied.

It is true that the proportion of different tumor types that are sequenced do not reflect the incidence of the specific cancers and to get a better estimate of the mutation rate across all cancers, there needs to be an adjustment to reflect this. However, there are several concerns with this approach and assumption that this is necessary.

(*)It does not appear as those the proposed reclassification scheme used for sequenced tumors as well as SEER data has been validated. It would strength the manuscript if there is evidence that the scheme that is used results in accurate classifications of tumors (from both SEER and from sequencing studies).

The issue of validation came up from multiple reviewers, and we agree this is an important point that we can better communicate. We have updated our manuscript to better communicate all of our validation methods. We will elaborate on all of our validation efforts and their communication here.

One type of validation could be whether our histological types are a valid abstraction of the ICD-O-3 codes. For this, our ROSETTA classifications should be fully valid because they are based upon the groupings within ICD-O-3 (i.e. ICD-O-3 categories are organized by similar forms of cancer, i.e. adenocarcinomas, squamous cell carcinomas). This is better communicated in our new manuscript, within our revised Supplementary Document 1, where in addition to a more thorough text description we also now have graphical diagrams that help communicate this point. In our revised manuscript, we now better communicate that we are not subdividing a histology or redefining it, and we better explain that ROSETTA is rather a series of groupings of previously-defined histologies (by the WHO). Our revised manuscript better explains that we are defining how these can be grouped in a manner consistent with the sequencing studies and WHO/ICD-O-3.

We cannot validate the histology of the samples, nor can we validate that the sequencing teams adhered to their stated form of cancer. We must assume that the sequencing teams are sequencing what they claim and what they describe in their meta-data. This is consistent with all genomics studies that are based upon analysis of available data, and is consistent with all interpretations of published genomic studies that implicitly assume the studies were histologically valid with respect to their samples.

Numerical validation is another form of validation that is also important. Our new supplementary figure (Figure S2) highlights our data analysis processes. Within this new figure we highlight various types of quality checks and validations (i.e. assuring

totals are conserved when a matrix is processed; having two or more independent implementations of a computational algorithm that obtain the same results, etc.). We also validate our ROSETTA processed SEER estimates by comparing our cancer proportions with those from the American Cancer Society on cancer incidence in the U.S. (i.e. Siegel et al, 2020) – and we report high agreement in the manuscript (Figure S1).

Within the revision, we have now better explained our methods in both the text and the supplementary materials. Within the main text, we have elaborated at each of the steps to communicate what we are doing and why. We have updated Supplementary Document 1 to make it much more interpretable and to be a more thorough walking through of the process.

Importantly, there is no “gold standard” to compare against for our final values. However, our analysis in Figure 2C,D,E where we compare our results against the TCGA pan-cancer analysis, shows improving correlation between the studies as more cancer samples were included in the previous TCGA pan-cancer analyses and as the correlation between number of samples per cancer type better correlated with epidemiological weighting. This is a reasonable form of final validation in the absence of gold standard. Additionally, we mention the work of Ian Prior and colleagues where they manually estimated overall RAS gene mutations rates across all cancers and how their conclusion with respect to epidemiologically weighted RAS genes is similar to ours, providing another source of external validation.

(*) The SEER data reports incident cancers whereas the sequencing data are not necessarily from incident cancers. It is not clear that adjusting rates obtained from both incident and prevalent cancers by rates of yields an accurate estimate.

(*) It is known that cancers that are sequenced differ from those that are not in meaningful ways and these differences may impact the mutation rates. Hence, merely adjusting to reflect the proportion of incident cancers may not yield accurate estimates of mutation rates across all cancers. One such difference is that sequenced tumors tend to be larger than those not sequenced and so they tend to reflect more advanced disease. There is evidence that mutation rates are different in advanced disease versus early disease. The proposed calculation cannot adjust for this. Hence, it is not clear that it truly is an accurate estimate. In addition, some of the sequenced tumors may have already been treated and the mutations may reflect mechanisms of drug resistance.

We agree these are all important consideration. Within our discussion, we have a section where we discuss issues that impact the quality of the estimate. For the revision, we have updated that text to mention the choice of tumor, which may be based on factors like tumor size, as a possible source of bias in the parent genomics studies. We have added the sentences “However, the decision of which tumors to sequence is likely influenced by other factors, such as tumor size and staging. Those choices and practices, which are beyond the scope of the present work, could introduce biases into

the data that are then reflected in the calculated mutation proportion estimates.” We also discuss alternative methods that may someday be possible, such as sequencing every incident cancer, which would result in no need for estimations. We also note here that tumor-focused studies, which are widely utilized to provide estimates for the rate of gene mutation within that tumor type, would have the same biases.

(*) Another concern is whether it truly matter what mutation rates across ALL cancers. For example, in SEER, most of the breast cancer cases are early stage due to screening, which captures cancers at an early time point. A significant portion of these cancers are cured with surgery and available treatments. Hence, does it matter what the mutation rate is in this cancer (at least from a treatment perspective)? This assumes that the mutation rates are the same in these cancers as in more advanced disease and in cancers for which there are no treatment options. Raises the question of whether pooling across all cancers really does make sense as well as whether it is accurate to pool across all cancers (when even within different subtypes of cancers, there are differing mutation rates, which is not captured in this calculation).

We agree that this cancer medicine is a complex area of medicine. We definitely appreciate nuance. Our lab publishes many papers that quantitatively explore cancer at a very detailed level, such as distinguishing between the effects of a KRAS G12D mutation from a G13D mutation and how it impacts the response to EGFR inhibition. Details definitely matter. But at the same time, general concepts and ideas can be very helpful for making sense of a complex discipline. The Hallmarks of Cancer is an example of one type of cancer simplification. Along the lines of simplification, there is also a natural human desire for knowing “the most”, “the best”, “the biggest”, etc. We address this desire by calculating the value, and we hope that our revised manuscript simultaneously highlights some of the complexity involved. Many, including those raised by the reviewers, we address directly; others are addressed indirectly by our highlighting that the values we present are estimates. We hope our emphasis on estimates and other factors will be useful for the readers who are less familiar with the complexity and for who the discussion of the complexity may lose them. It is currently very common for cancer biologists to cite the rate of mutation to a gene as observed in a focused study of a cancer type (i.e. TCGA lung adenocarcinoma) or of a pan-cancer analysis (i.e. a cBioPortal all-studies query) as representing rates in individual and all cancers; we believe that our manuscript is both improving upon these common practices while also highlighting the assumptions and possible biases in the data.

Additionally, the discussion of the NCI RAS initiative that was heavily motivated by the perceived 30% or more of all cancers having RAS genes mutate is an example where an incorrect perception may have resulted in an increased federal research investment than would have been made had the true proportion been known.

(*) Assuming that non-represented cancers have the same mutation rates as the other cancers is questionable. It is known that tumor burden (as measured by the

number of mutations within a tumor) differs by cancer.

We completely appreciate this point. We will first note that practically speaking, many people in the field currently implicitly consider the percentage of samples with a mutation from cBioPortal, TCGA, AACR Genie, etc. as representing the epidemiologically weighted average when they report such values as a metric for the prevalence of that mutation. In our experience sharing our work with colleagues for feedback, they had never thought deeply about the assumption they were making when they utilized those data as they had assumed that the data were representative. As all cancer types have not been sequenced, the assumption raised above is the common practice in the field.

That said, we spent a lot of time thinking about how to handle non-sequenced cancer types. This was a major topic of discussion at the beginning of our study, before we knew what proportion of cancers we would have sequencing data for and before we knew that mutations would be found at a lower rate than we had expected based on conventional wisdom. We thought about how we might be able to obtain better estimates than the upper bound (assuming all un-sequenced cancers have the mutation), the lower bound (assuming none of the sequenced cancers have the mutation), and the “sequenced-cancers approximate unsequenced cancers” approximation. On seeing our high coverage (93% of cancer types), the low rates of mutation overall, and the variation between cancer types, we felt much more comfortable with the “sequenced cancers approximate unsequenced cancers” estimate.

Within the revised manuscript, we have rephrased our description of this assumption/estimation so that it is more clear. We also highlight that our text is very explicit that the values we calculate and present are estimates. We disclose our methods and assumptions, and we have added confidence intervals to our estimates (methodology discussed in text and in response to reviewer who requested). We welcome efforts to build upon our work, and we are sharing our code so others can adapt and extend the approach.

Reviewer #2 (Remarks to the Author): Expert in cancer genomics and bioinformatics

This is an elegant study tackling an important unresolved question: how do the frequency of somatic driver mutations reported by large pan-cancer consortia compare to that in the broad population of cancers. To address it, the authors developed a mapping strategy they call ROSETTA, then apply it to two versions of TCGA pan-cancer data. The study is generally well-written, and admirably documented with robust and very valuable supplementary.

We thank Reviewer #2 for the kind words and for the helpful review of our manuscript. The revised manuscript includes many changes and updates to address the contents of the review, as detailed below.

1. Terminology

An absolutely critical error throughout is the use of the term "mutation". The authors appear to only be analyzing one of the several classes of mutations -- "somatic functional single nucleotide variants (SNVs)". Almost every conclusion in this paper is incorrect when phrased generally as mutation because of the large number of translocations, inversions, gains, deletions and so forth. It is fine in colloquial terminology to speak of "driver mutations" when only referencing SNVs, but in the literature this would be a critical error. The entire paper absolutely needs to be updated to use the technically correct terminology, including a title change.

We thank reviewer #2 for requesting this important clarification. We have updated the manuscript to more clearly communicate that we are focusing on exome sequencing studies and therefore focus on exonic mutations that impact coding (missense, nonsense, and indels) and that we are not including intronic mutations or fusions. Nor are we calling copy number variation (which can be called with exomic data – but inconsistently across cancers for factors such as a varying proportion of cancerous to non-cancerous cells within a tumor). We put this important clarification in a new paragraph within the genomics section where we explicitly define the types of mutations upon which we here focus. The paragraph also explains why we do not investigate other forms of mutation in this study. The new paragraph also highlights that our ROSETTA approach could be applied to other types of mutation. We also make it clear that when we use the word “mutation”

2. Gene Symbols

Supplementary tables show gene symbols like gene symbols of 1-Dec and 3-Mar that appear to be artifacts of Excel processing

Thank you for catching these. We have updated our Excel formatting so that such changes should not happen again.

3. ROSETTA

Significantly more information is required on ROSETTA:

a) How was it validated? Please attach estimates to the error rate of the mapping process.

Updates on validation are also discussed in the response to Reviewer #1, who also requested more information and validation. To briefly summarize here, we have provided more thorough explanation of ROSETTA and its validation in the text, as well as in a revised Supplementary Document 1 and a new Supplementary Figure 2, which highlights data integrity checks and code validation. Within the text, we better explain that ROSETTA is largely a regrouping to higher levels of granularity in accordance with

ICD-O-3, thus ROSETTA category is valid in the sense ICD-O-3 is valid. We also better explain our numerical and computational validation within the text and within our figures.

b) Where is there a list of all TCGA samples with their original IDs and new ROSETTA classifications?

We have updated Supplementary Table 2, which lists all genomic studies, to also include which ROSETTA classifications samples have been mapped to. Additionally, within our supplementary code, we have the files that specify the ROSETTA code for each of the >19,000 samples incorporated into this study.

c) Please provide a software implementation or equivalent that could be used to map datasets beyond TCGA (e.g. PCAWG or GENIE)

We are providing all of our software that is needed for reproducing our study as part of the supplementary materials. We cannot provide a full implementation of PCAWG or GENIE at this time because of the manual curation steps that are required. Our improved description of ROSETTA, of the process of mapping to ROSETTA, and the inclusion of our code, however, would allow others to implement the same process.

4. Visualization

Please ensure scatterplot axes are exactly matched in their range and tick-marks, and include $y=x$ lines to allow direct comparison

We have made the requested changes to the figures and figure legends, and agree this makes the figures more useful and help guide the interpretation better. Thank you.

5. Statistics

a) It is unclear in cases like the adenocarcinoma vs. squamous cell carcinoma if the inferred differences are actually statistically significant. One presumes so, but it is difficult to ascertain.

b) Similarly please attach confidence intervals to population driver estimates based on the accuracy of estimating driver frequencies from a few hundred cases per cancer type.

We have added confidence intervals for all of our bar plots and for all of our supplementary tables. Within our adenocarcinoma vs. squamous cell carcinoma comparisons the scatter plot would be too busy to show all of the confidence intervals. All confidence intervals for these data are in supplementary table 4, and the confidence intervals are non-overlapping for the genes upon which we focus (and many, many, more). Our figure legend has been updated. We have also updated our methods to describe our statistical methods.

Reviewer #3 (Remarks to the Author): Expert in epidemiology and statistics, reviewed with Reviewer #4

The authors generate mutation rates among all cancers within the US population. They note that in prior works with COSMIC, TCGA, GENIE, the representation of difference cancers in the resources is not provided and is not designed to be a proportional relative to those cancers in a population. The authors develop a technique they call ROSETTA (Reclassification Of Sequencing and Epidemiological Tumor Type Annotations), which appears to interconvert between the different nomenclatures used to label cancer diagnoses in the SEER database and several genomic studies from cBioPortal. The authors show that certain genes are commonly mutated (eg p53); but there is a long tail, and there are many other mutations that may be present.

Overall, this is interesting work. The writing is generally clear. There are almost no typos.

The authors should be commended for trying to do this large-scale analysis, and I hope my comments are helpful.

We thank Reviewer #3 for reading our work and providing this helpful review. We have made many changes to the manuscript in accordance with the requests in the review, detailed below.

Major comments:

(1) I think the methods need to be clarified, with more details of each step, and more description in the text and figures. I reviewed this manuscript in detail with a computational biostatistician who has knowledge of the genomic databases, to complement my own understanding of SEER, and we think the authors are doing the following, but we need more clarification. I'm going to summarize the steps we think the authors are using, and provide questions within each step:

We appreciate Reviewer #3 and #4 for taking the time to work carefully through our approach. We will go through each step as they outline it below.

Their point about needing additional information is well-taken by the authors. To address, we have (a) completely revised our Supplementary Document 1 to better communicate our processing in detail and with illustrations; (b) added a new supplementary figure (Figure S2) that shows the processing at a high level while also indicating key quality control and validation steps; (c) extended our descriptions of methods and processing in the main text and its methods section. These changes address the specific questions raised by Reviewer #3 and Reviewer #4, as well as #1 and #2.

If I understand correctly, in step 1 entries in the table are observed counts based on SEER data. How did the authors come up with the groupings in Sup Document 1?

We have revised our main text and our Supplementary Document 1 to better communicate our ROSETTA reclassifications. Importantly, we better communicate that the ROSETTA groupings take advantage of higher-level groupings of the WHO ICD-O-3 categories upon which they are based. We also better communicate how choices are pragmatic, based upon the types of cancer that have received exome sequencing, including specific examples in Supplementary Document 1 that describe such cases (cholangiocarcinoma; pancreatic neuroendocrine; pancreatic acinar cell; uveal melanoma; acral lentiginous melanoma).

How are histologies besides those listed included in the current work? It appears, based on Sup File 1 and Figure 4 that most of the histologies are distilled to adeno vs SCC vs melanoma vs TCC. However, is there a reason this figure doesn't include small round blue cell tumors, gliomas, lymphomas?

This section was unclear, so we have added a new paragraph to the beginning of our pan-subtype analysis to better communicate this information. Figure 4A highlights that the four we drilled down on are not comprehensive but do comprise a majority of cancers. Of note, we do have gliomas and lymphomas that were sequenced (all of the ROSETTA categories and their weighting of all cancers are listed in Supplementary Table 1). Gliomas, lymphomas, small round blue cell tumors, etc. would all be in the "All other tumors" gray wedge of the pie chart.

The four categories drilled down on in Figure 4 are simply the major categories, consistent with ICD-O-3, that were most common (and for which we also had good data.) The four sub-histology lists are among the most commonly represented coarse-grained categories covering 83% of the cancer diagnoses. In the supplementary code included, we have provided a function to generate similar data for any set of sub-histologies of interest.

I wonder if the authors could better summarize Sup File 1 to provide a frequency table of the codes, and then provide a figure showing how the histology codes are grouped and the rationale for grouping.

We have improved, significantly updated, and illustrated Supplementary Document 1 to better explain the regrouping. The frequency of each specific histology code is presented in Table S1. The rationale for grouping is better explained in the main text and in the new Supplementary Document 1. All of our code needed to reproduce our work is also provided as a Jupyter notebook.

In step 2, it looks like they use the ROSETTA class scheme to create matrix vs the anatomical location. The entries are estimated counts based on entries from Step 1. My comments for this step are:

The interpretation above is correct.

Why use “site” in step 1, and use “anatomical location” in step 2?

We agree that we used the terms interchangeably in the text as synonyms. We have updated our text for clarity. Importantly, SEER ICD-O-3 codes include a site/anatomical location/topography code, and cancer genomic studies also commonly have an anatomical location that may be less formally defined than the SEER ICD-O-3 code. Our updated manuscript and accompanying supplementary material should better communicate the nuance of the ROSETTA method and implementation.

How do the authors account for multiple entries within each column or row? eg, if within lung cancer, there is non small cell lung cancer, adeno, squamous cell, small cell, how are these subdivided?

We thank the reviewer for this helpful comment, as this motivated the new illustrations in our supplementary document. Different forms of lung cancer do get mapped to different ROSETTA codes; we have one for small cell, squamous, and adeno; additionally, other tumors that appear in the lung (lymphoma, mesothelioma) will receive their own ROSETTA code. We believe that Supplementary Document 1 and Diagram 2 from that document will help show that one location (i.e. lung) can have many different cancers mapped to it. This specific question by the reviewer was very helpful for pinpointing how we can better communicate our approach.

In Step 3, they bring in mutation data, creating a matrix of genes vs ROSETTA classes. This seems clear, but I think I need to understand the prior steps better.

This is correct. We hope that our new work better explains the prior steps; we are glad this part was clear.

In Step 4, the ROSETTA classes and genes are converted to percentages. Entries are column frequencies based on counts of the matrix in Step 3. This step depends on having mutation data and SEER data. For certain cancers, it sounds like it is possible that there are values of 0, if there were no genomic data for that cancer. How do the authors account for this? How often does it occur?

We better clarify our approach in the text. We had sequencing data for 93% of cancer cases. However, that means we have no representative sequencing data for 7% of the cancer cases that occur in the US. Our approach is that we assume that the 93% are a good estimate of the mutation proportions for the 100% (i.e. that the missing 7% will be well approximated by the first 7%). We have added and rephrased our text to make this more obvious and to address how it may impact calculation in order to address this request from Reviewers #1, #3, and #4. Additional discussion of this approximation is included in the response to Reviewer #1.

In Step 5, in each ROSETTA class, they look at the total number of SEER samples,

based on Step 2. There might be ROSETTA data for cancers we don't have genetic data, producing a value of 0. The data are scaled proportionately. The n are converted to percentages.

Step 6. Take the genes (in rows) vs classes of ROSETTA (in columns), and multiply this by ROSETTA classes, to provide a weighted gene mutation frequency. Is this understanding correct?

Correct on Step 5 and Step 6.

(2) Comments about tables and figures:

Table 1 is too difficult to understand with so much text. Could this be transformed to be a figure?

We have simplified Table 1 by focusing on less genes and by removing the proportion of cancers that were adenocarcinoma. (Importantly, all genes and their overall proportion mutated are in Supplementary Table 3, and the adenocarcinoma data, along with data on squamous cell carcinoma, melanoma, and transitional cell carcinoma, is in Supplementary Table 4.) We hope that this table is less difficult to understand with cleaner presentation and better focus on key information we wanted to communicate. We have also added a table legend that better explains the included information.

Figure 1: how are patients with multiple mutations included in this work?

We have updated the text in our section "Reclassification of genomic studies by ROSETTA and pooled analysis" to better communicate that we count a patient with multiple mutations as a single count (i.e. the patient has the gene mutated one or more times (1) or not (0)).

Figure 3: it looks like there is a long tail of mutations, and mutation frequency of any gene is ~3%. Could it be that these mutations are normal and expected? Or are they all expected to cause pathology?

We have updated our description of our data that is presented in Figure 3 to better highlight that we are counting all mutations, and not only those known to cause pathology, and that our prevalence rates are therefore overestimates for driver frequency. For example, going with Figure 3A we have added "As our analysis includes both driver and passenger mutations and is simply looking at the total number of mutations found within the genes in the Cancer Gene Census, this suggests that the overall occurrence of driver gene mutations in human cancer is even less than the values we present here."

In the paragraph describing Figure 3B, we state "Again, we note that these values are for all mutations and are not limited to those likely to be pathogenic – thus, these values will overestimate the frequency with which potentially targetable pathogenic mutations will be found in these kinases."

Supplementary Table lists all Consensus Cancer Genes (>500), all kinases (>500), and all RAS pathway genes (>200), and there are many genes with less than ~3% mutated. We apologize that our presentation of the top 50 of each class in Figure 3, which ended at ~3%, gave a different impression. We have updated our figure legend to clarify the additional genes within each class are included in supplementary table 3.

I am having a difficult time understanding supplementary file 1

With regard to Supplementary Document 1, we have replaced the previous word file that listed a complex dependency on location and histology codes to a more readable document that explains the philosophy behind ROSETTA with illustrations that communicate how the groupings apply to epidemiological and genomic data.

We are also including a supplementary zip file that includes code to reproduce our study. For the revision, we are providing a Jupyter-notebook based set of Python scripts that reproduces our analysis. This can be downloaded and used to process the input matrices and generate the same calculated output values that we present. Additionally, code is provided to download and process the genomic data.

The previous contents of the earlier version of Supplementary Document 1 are available in the supplementary code zip file.

(3) Minor comments:
Carcinoma misspelled in Figure 4D

We have corrected this. Thank you for catching this and bringing it to our attention.

Reviewer #4 (Remarks to the Author): Expert in epidemiology and statistics, reviewed with Reviewer #3

Key Results

The authors have developed ROSETTA, a novel method for classifying tumor types that enables them to match epidemiologic data from SEER with cancer genomic data from the CBioPortal. Using ROSETTA, they compute population-based estimates of gene mutation frequencies across all cancers in the U.S. as well as specific types of cancer, e.g., all squamous cell carcinomas. These population-based mutation frequencies are then compared to gene mutation frequencies obtained in two TCGA “PanCan” studies as well as other previous publications. The findings suggest that in the general population, mutations in cancer affect a few

driver genes with high frequency as well as a large number of genes with low frequency (long tail). Specific findings include a high frequency of TP53 mutations, a lower than expected frequency of KRAS mutations, and common mutations of epigenetic modifiers KMT2C, KMT2C, and ARID1A.

We thank Reviewer #4 for reading our work and providing this helpful review. We have made many changes to our manuscript to address the points raised by Reviewer #4, which are detailed below.

Validity

I have no concerns.

Significance

As the authors note, ROSETTA represents one possible classification scheme that could be used to merge epidemiologic and genomic data sets, and because of this the mutation frequencies presented in the paper should not be viewed as definitive. Regardless, ROSETTA is a principled and well-developed method, and the fact that the authors have applied it to obtain population-level gene mutation frequencies using data from SEER and high-profile cancer genomics studies is noteworthy and likely novel.

Data and Methodology

The authors utilize publicly available data from SEER and the CBioPortal, both of which are widely used. Thus there are no concerns about data quality. The approach detailed in the Methods section produces the desired weighted gene mutation frequencies.

Analytical Approach

See the above comments regarding Methodology.

We thank Reviewer #4 for these comments, and are glad to hear there are no concerns regarding data quality and validity.

Suggested Improvements

- While it is natural to reference the two TCGA PanCan papers, the authors use the terms “not well estimated” and “large errors” when comparing their population-based mutation frequencies with the mutation frequencies in these studies. This language suggests that the mutation frequencies from the PanCan studies should be directly comparable to the population-based mutation frequencies computed by the authors, which is not the case. Please clarify.

We have rephrased these sections to remove those specific phrases and to make it clear that authors of those manuscripts were not attempting to generate epidemiologically weighted estimates. We have rephrased that section to state that one would be making an error to assume that the TCGA data were effectively epidemiologically weighted estimates (which is a common error we have witnessed

many people make). We thank the reviewer for pointing out this point, and we hope our rephrased section addresses this matter satisfactorily.

- The authors note that the high frequency of TTN and OBSCN mutations may arise because of the large sizes of these genes. With that in mind, it would be useful to know whether these mutations are often pathogenic or simply passenger mutations.

We have added additional text and a reference to highlight that these proteins are not believed to be pathogenic, while also including a reference that suggests there may be driver gene function to some of these variants. Additionally, we contrast their rate with the lesser rate of known drivers and the implication for cancer driver gene selection forces.

- Related to the previous comment, it would be interesting to know how much the population gene mutation frequencies change if you restrict to pathogenic mutations. Perhaps this can be computed readily after filtering the MAFs.

We definitely agree this would be interesting, but we believe this would be outside of the scope of the present work. We highlight that the ROSETTA approach can be put to many uses, and a focus on known pathogenic drivers, targetable drivers, etc., would be one of interest. We also believe that ROSETTA reclassified and epidemiologically weighted mutation data could be useful for efforts to identify new cancer driver genes, and we have added text to mention other uses for ROSETTA in the discussion.

- The lower than expected frequency of KRAS mutations is noted in the paper. Can you provide any insight into why this occurs? Presumably there are specific tumor types that are common in the population yet have low RAS mutation frequencies, but I don't think this is ever explicitly stated.

We have now elaborated upon the reasons in the manuscript, including highlighting the very common cancers (breast and prostate) that rarely have RAS mutations. We believe this section is now more clear and reads better. Thank you for this suggestion.

- The phrase "Cancer sequencing data, once analyzed by a variety of statistical approaches" suggests that statistically-based methods are no longer applied when analyzing sequencing data. This is not the case, as evidenced by the widespread usage of MutSig and other related methods. Please clarify this point.

The interpreted suggested meaning was not our intention; we have rephrased this sentence to better communicate the intended message, which was that there are many driver genes in the RAS pathway and that cancer genomic efforts have led to the discovery of even more RAS pathway drivers.

- The authors note that genes with mutation frequencies above 8% were either on the Consensus Cancer Gene Census list or were very large proteins. How was the 8% cutoff chosen? Also, only a handful of genes satisfy this criteria, so perhaps they

should be listed in the text.

This was an arbitrary value, meant for us to communicate that even by inspection there are no high frequency, missing, driver genes. The definition of “high frequency” itself is up for debate – we are thinking on the scale of the other common drivers. We removed that sentence –this manuscript is not a “driver gene discovery” manuscript, and there are no obvious, high-frequency, surprises to describe. Thus, we hope the removal improves the manuscript by giving it a better focus on our results.

- It looks like there is considerable overlap between the percentages in Table I and the percentages shown in Figure 3A and Figure 4D. Perhaps this table could be moved to the Supplement.

We have updated Table I by removing the data that was in Figure 4D (which is also a subset of supplementary table 4). We now better use our space to focus on our originally intended message, which was to focus on the three cancer histologies that account for the most cases of cancer with these high-frequency cancer driver genes mutated.

- In the Methods section the authors refer to the $m \times n$ matrix C and the $k \times 1$ matrix S. Why not specify the values of m, n, and k?

We have updated our methods to list the values of m,n,and k, while also keeping the m,n,k notation because the approach is adaptable. Thank you for this suggestion.

- Not all readers will be familiar with matrix multiplication or how it is being used in the current context to compute population-level mutation frequencies. Because of the large number of ROSETTA classifications, I recognize that it may not be feasible to show a real worked example. However, even a toy example would be useful. Perhaps this could be added to the supplement.

We have extended our supplementary document to include a simple example that shows how the frequencies combine for these estimates. We have included diagrams, too, to communicate the matrix multiplication with a toy example. Thank you for the suggestion.

- The discussion of gene mutation frequencies in melanoma and transitional cell carcinoma seems brief in comparison to the discussion of mutation frequencies in adenocarcinoma and squamous cell carcinoma. For example, the authors note that TP53 is not the most commonly mutated gene and PIK3CA is not the most commonly mutated oncogene in melanoma without actually naming the most commonly mutated gene or most commonly mutated oncogene. Admittedly this information is presented in Figure 4, but it should appear in the text as well.

We have added additional text to complement Figure 4. Thank you for the suggestion.

Clarity and Context

I have no concerns.

References

I have no concerns.

My Expertise

Although I have worked with SEER data in the past, my role was primarily that of a data analyst. As such, I am not well-versed in the cancer epidemiology literature, and thus I cannot assess whether similar studies have been performed and should be referenced.

Thank you for contributing to the review and for the helpful suggestions.

REVIEWERS' COMMENTS

Reviewer #1 (Remarks to the Author):

The authors have responded satisfactorily to my comments.

Reviewer #2 (Remarks to the Author):

The authors have fully addressed my concerns. My two final suggestions are:

a) add a supplementary file that more clearly lays out the step-by-step procedure required to do the ROSETTA mapping to a new dataset, along with the appropriate links to code and sufficient manual step details. The procedure is currently split across too many places to be reusable

b) highlight in the text the filename of the TCGA-Rosetta code mapping file (which is critical for data-reuse)

Reviewer #3 (Remarks to the Author):

This looks good, my comments have been addressed.

Reviewer #4 (Remarks to the Author):

I commend the authors for their work. Based on the explanations in the "Response to Reviewers" document and the changes made to the text, figures, and supplement, I have no additional suggestions or corrections. Therefore, I recommend that the manuscript of Mendiratta et al. be published in Nature Communications.

Response to Reviewers

REVIEWERS' COMMENTS

Reviewer #1 (Remarks to the Author):

The authors have responded satisfactorily to my comments.

RESPONSE: We appreciate Reviewer #1's help with the review process, and our manuscript is better for addressing Reviewer #1's previous suggestions.

Reviewer #2 (Remarks to the Author):

The authors have fully addressed my concerns. My two final suggestions are:

a) add a supplementary file that more clearly lays out the step-by-step procedure required to do the ROSETTA mapping to a new dataset, along with the appropriate links to code and sufficient manual step details. The procedure is currently split across too many places to be reusable

RESPONSE: All of the software needed for reuse is now available in a single zipped folder. We have had non-computational lab members download the software and implement it. We have added a new section on implementation to our Supplementary Methods. This section provides links to code and descriptions of where to find the specific files. The supplementary methods is quite thorough with regards to the manual process required to implement ROSETTA.

b) highlight in the text the filename of the TCGA-Rosetta code mapping file (which is critical for data-reuse)

RESPONSE: For each genomics study, there is one text file that provides mappings between the individual cases within a genomics study and the assigned ROSETTA code(s). We have updated our Supplementary Methods to describe where each of these files is located in our supplementary software. Supplementary Table II lists each genomics study and which ROSETTA codes are utilized for each study. We appreciate Reviewer #2's help with the review process, and our manuscript is better for addressing Reviewer #2's present and previous suggestions.

Reviewer #3 (Remarks to the Author):

This looks good, my comments have been addressed.

RESPONSE: We appreciate Reviewer #3's help with the review process, and our manuscript is better for addressing the Reviewer #3's previous suggestions.

Reviewer #4 (Remarks to the Author):

I commend the authors for their work. Based on the explanations in the "Response to Reviewers" document and the changes made to the text, figures, and supplement, I have no additional suggestions or corrections. Therefore, I recommend that the manuscript of Mendiratta et al. be published in Nature Communications.

RESPONSE: We appreciate Reviewer #4's help with the review process, and our manuscript is better for addressing Reviewer #4's previous suggestions.